# Online Robust Policy Learning in the Presence of Unknown Adversaries

Aaron J. Havens, Zhanhong Jiang, Soumik Sarkar

Department of Mechanical Engineering
Iowa State University
Ames, IA 50011
{ajhavens,zhjiang,soumiks}@iastate.edu

## Abstract

The growing prospect of deep reinforcement learning (DRL) being used in cyber-physical systems has raised concerns around safety and robustness of autonomous agents. Recent work on generating adversarial attacks have shown that it is computationally feasible for a bad actor to fool a DRL policy into behaving sub optimally. Although certain adversarial attacks with specific attack models have been addressed, most studies are only interested in off-line optimization in the data space (e.g., example fitting, distillation). This paper introduces a Meta-Learned Advantage Hierarchy (MLAH) framework that is attack model-agnostic and more suited to reinforcement learning, via handling the attacks in the decision space (as opposed to data space) and directly mitigating learned bias introduced by the adversary. In MLAH, we learn separate sub-policies (nominal and adversarial) in an online manner, as guided by a supervisory master agent that detects the presence of the adversary by leveraging the *advantage* function for the sub-policies. We demonstrate that the proposed algorithm enables policy learning with significantly lower bias as compared to the state-of-the-art policy learning approaches even in the presence of heavy state information attacks. We present algorithm analysis and simulation results using popular OpenAI Gym environments.

## 1  Introduction

Real applications of cyber-physical systems that utilize learning techniques are already abundant such as smart buildings [1], intelligent transportation networks [2], and intelligent surveillance and reconnaissance [3]. In such systems, Reinforcement Learning (RL) [4, 5] is becoming a more attractive formulation for control of complex and highly non-linear systems. The application of Deep Learning (DL) has pushed recent advances in RL, namely Deep RL (DRL) [6, 7, 8]. Particularly in 3D continuous control tasks, DL is an indispensable tool due to its ability to generalize high dimensional state-action spaces in Policy Optimization algorithms [9], [10]. Notable variance reduction and trust-region optimization strategies have only furthered the performance and stability of DRL controllers [11].

Although DL is generally useful for these control problems, DL has inherent vulnerabilities in the way that even very small perturbations in state inputs can result in significant loss in policy learning performance. This becomes a very reasonable cause for concern when contemplating DRL controllers in real-world tasks where there exist, not only environmental uncertainty, but perhaps adversarial actors that aims to fool a DRL agent into making a sub-optimal decision. During policy learning, information perturbation can be generally thought of as a bias that can prevent the the agent from effectively learning the desired policy. Previous attempts in mitigating adversarial attacks have been successful against specific attack models, however, such robust training strategies are typically off-line (e.g., using augmented datasets [12]) and may fail to adapt to different attacker strategies in an online fashion. Recently [13] has taken a model-agnostic approach by predicting future states, however it may be susceptible to multiple consecutive attacks.

**Contributions**: In this paper, we consider a policy learning problem where there are periods of adversarial attacks (via corrupting state inputs) when the agent is continuously learning in its environment. Our main objective is online mitigation of the bias introduced into the nominal

Table 1: Comparisons with different robust adversarial RL methods

| Method | Online | Adaptive | Attack-model agnostic | Mitigation |
|---|---|---|---|---|
| VFAS [13] | ✓ | ✗ | ✓ | ✓ |
| ARDL [12] | ✗ | ✗ | ✗ | ✓ |
| MLAH [This paper] | ✓ | ✓ | ✓ | ✓ |

Online: no offline training/retraining required, Adaptive: can adapt to a change in attack strategy, Attack-model agnostic: assumes no specific attack model, Mitigation : is the impact of the attack actively mitigated?

policy by the attack. We only consider how an attack affects the return instead of optimizing the observation space. In this context, our specific contributions are:

1. **Algorithm** We propose a new hierarchal meta-learning framework, MLAH that can effectively detect and mitigate the impacts of adversarial state information attacks in a attack-model agnostic manner, using only the advantage observation.
2. **Analysis**: Based on a temporal expectation definition, we analyze the performance of a single mapping policy and our proposed multi-policy mapping. Visitation frequency estimates leads us to obtaining a new pessimistic lower bound for TRPO and variants.
3. **Implementation**: We implement the framework in widely utilized Gym benchmarks [14]. It is shown that MLAH is able to learn minimally biased polices under frequent attacks by learning to identify the adversaries presence in the return.

Although we mention several relevant techniques on learning with adversaries, we only contrast methodologies in table 1 that aim to mitigate adversarial attacks, as other papers [15], [16] do not claim to do so. We compare our results with the state-of-the-art PPO [17] that is sufficiently robust to uncertainties to understand the gain from multi-policy mapping. The source code is available on https://github.com/AaronHavens/safe_rl.

**Related work**: Attacks on deep neural networks and mitigation strategies have only recently been studied primarily for supervised classification problems. These attacks are most commonly formulated as first order gradient-based attacks, first seen as FGSM by Goodfellow et al [18]. These gradient based perturbation attacks have proven to be effective in misclassification, with the corrupted input often being indistinguishable from the original. The same principle applies to DRL agents, which can drastically affect the agent performance and bias the policy learning process. The authors in [19] showed a threat model that considered adversaries capable of dramatically degrading performance even with small adversarial perturbations without human perception. Three new attacks for different distance metrics were introduced in [20] in finding adversarial examples on defensively distilled networks. The authors in [21] introduced three new dimensions about adversarial attacks and used the policy's value function as a guide for when to inject perturbations. Interestingly, it has been seen that training DRL agents on designed adversarial perturbations can improve robustness against general model uncertainties [16], [15]. The adversarial robust policy learning algorithm [22] was introduced to leverage active computation of physically-plausible adversarial examples in the training period to enable robust performance with either random or adversarial input perturbations. Another robust DRL algorithm, EPOpt-$\epsilon$ for robust policy search algorithm [23] was proposed to find a robust policy using the source distribution. Note that the recently mentioned methods do not aim to mitigate adversarial attacks at all, but intentionally bias the agent to perform better for model uncertainties.

## 2 Preliminaries and Problem Formulation

In this paper, we consider a finite-horizon discounted Markov decision processes (MDP), where each MDP $m_i$ is defined by a tuple $M = (\mathcal{S}, \mathcal{A}, \mathcal{P}, r, \gamma, \rho_0)$ where $\mathcal{S}$ is a finite set of states, $\mathcal{A}$ is a finite set of actions, $\mathcal{P}$ is a mapping function that signifies the transition probability distribution, i.e., $\mathcal{S} \times \mathcal{A} \times \mathcal{S} \to \mathbb{R}$, $r$ is a reward function $\mathcal{S} \to \mathbb{R}$ with respect to a given state and $r \in [r_{min}, r_{max}]$, $\rho_0$ is a distribution of the initial states and $\gamma \in (0,1)$ is the discounted factor. The finite-horizon expected discounted reward $\mathcal{R}(\pi)$ following a *policy* $\pi$ is defined as follows:

$$\mathcal{R}(\pi) = \mathbb{E}_{s_0, a_0, \dots} \left[ \sum_{t=0}^{T} \gamma^t r(s_t) \right] \tag{1}$$

where $s_0 \sim \rho_0(s_0), a_t \sim \pi_i(a_t|s_t), s_{t+1} \sim \mathcal{P}(s_{t+1}|s_t, a_t)$. We want to maximize this discounted reward sum by optimizing a policy $\pi : \mathcal{S} \to \mathcal{A}$ map, discussed next.

### 2.1 Trust Region Optimization for Parameterized Policies

For more complex 3D control problems, policy optimization has been proven to be the state-of-the-art approach. A multi-step policy optimization scheme presented in [11] dually maximizes the improvement (Advantage function) of the new policy while penalizing the change between the old

and new policy described by a statistical distance, namely the Kullback Liebler divergence. For continuous control policy optimization a variant of the advantage function is often used being the *Generalized Advantage Function* (GAE) from [24], which is parameterized by $\gamma$ and $\lambda$ where $V(s_t)$ is the value function. Intuitively, GAE attempts to balance the trade-off between bias and variance in the advantage estimate by introducing the controlled parameter $\lambda$. We will use this in policy optimization as well as a method for temporal state abstraction later in the proposed algorithm.

$$A_{GAE,t} = \zeta_t + (\gamma\lambda)\zeta_{t+1} + ... + ...(\gamma\lambda)^{T-t+1}\zeta_{T-1} \tag{2}$$

where $\zeta_t = r_t + \gamma V(s_{t+1}) - V(s_t), \gamma, \lambda \in [0,1]$.

## 2.2 Meta-Learned Hierarchies

As a basis for our proposed MLAH framework, we consider a task with multiple objectives or latent states. In this context, we define a finite set of MDPs $\mathcal{M}:\{m_0, m_{,1}, \cdots, m_n\}$, where an MDP $m_i$, $i \in \{0, 1, \cdots, n\}$ is sampled for learning at time $t$. There exists a set of corresponding sub-policies $\Pi$ : $\{\pi_0, \pi_1, \cdots, \pi_n\}$ which may individually be used at any instant. We then have $\mathcal{M} \rightarrow \mathcal{R}$ and define a joint hierarchal objective for $\mathcal{M}$ composed of sub-policies:

$$\mathcal{R}(\Pi) = \mathbb{E}_{s_0,\pi_0,m_0...}\left[\sum_{t=0}^{T}\gamma^t r(s_t)|\, m_i, \pi_i\right] \tag{3}$$

Every $m_i$ can be thought of as a unique objective in the same state-action space. In our case, the RL agent is not aware of the specific $m_i$ at time $t$. This could alternatively be thought of as a partially observable MDP (POMDP), however in this work we introduce a hierarchal RL architecture to explain the latent state. This hierarchal framework depicted in Figure 1 has been presented in [25] as *Meta-Learned Shared Hierarchies* (MLSH). $\pi_{master}$ de-

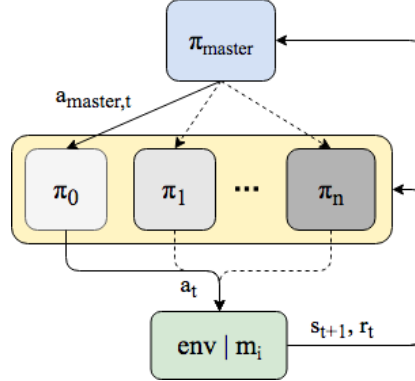

Figure 1: A meta learning hierarchy similar to MLSH in [25]. The master is tasked with choosing a sub policy to maximize return in the current MDP $m_i$.

scribes an agent who's task is to select the appropriate sub-policy to maximize return. The master policy, $\pi_{master}$ receives the observed reward and environment state. This mapping is far easier to learn as apposed to re-learning each sub policy which may be re-used. Since each $m_i \in \mathcal{M}$ has a different $\mathcal{S} \rightarrow \mathcal{R}$ mapping, this makes $\pi_{master}$ have a non-stationary mapping across $\mathcal{S}$ which requires the parameters of $\pi_{master}$ to be reset on a predetermined interval.

## 2.3 Adversary Models

We consider adversaries that perturb the state provided to the agent at any given time instant. Formally,
**Definition 1.** *An adversarial attack is any possible state observation perturbation that leads the agent into incurring a sub-optimal return, which is less than the return of the learned optimal policy. In other words, $\mathcal{R}(\pi|attack) < \mathcal{R}(\pi)$. The adversary **may only** perturb the state observation channel, and not the reward channel itself.*
Note, when discussing adversarial attacks, a common practice is to mathematically define a feasible perturbation with respect to the observation space. This work presents an alternative approach (later in the analysis Section 4) by focusing on expected frequency of attacks only and how it realizes in the RL decision space. This results in a framework which is more agnostic to a specific attack-model and considers more than just the observation (data) space. However, it is important to note that the RL agent is not aware of any attack-model specifications.

# 3 Proposed Algorithms

We begin with a brief motivation to the proposed Meta-Learned Advantage Hierarchy (MLAH) algorithm. An intelligent agent, such as a human with a set of skills, when presented with a new task, should try out one of the known skills or policies and examine its effectiveness. When the task changes, based on the expectation of usefulness of that skill, the agent may keep using the same skill or try another skill that may seem most appropriate for that task. In this context, given that the agent has developed accurate expectations of its sub-policies (skills), if the underlying task were to change at anytime, the agent may notice that the result of its action has changed with respect to what was expected. In an RL framework, comparing the expected return of a state to the observed return of some action is typically known as the *advantage*. Therefore, such an advantage estimate can serve as a good indicator of underlying changes in a task that can be leveraged to switch from one sub-policy to another more appropriate sub-policy.

With this motivation, we can map the current problem of learning policy under intermittent adversarial perturbations as a meta-learning problem. As our adversarial attacks (by definition 1) create a different state-reward map, a master policy may be able to detect an attack and help choose an appropriate sub-policy that corresponds to the adversarial scenario. More formally, we begin with two random policies that are meant to represent the two distinct partially observable conditions in our MDP, nominal states and adversarial perturbed states. One may begin by pre-training $\pi_{nom}$ in isolation seeing only nominal experiences. Since we can not assume or simulate the adversary, typically it is not possible to pre-train $\pi_{adv}$ and it must be left to $\pi_{master}$ to identify this alternative mapping. For each episode, we begin by collecting a trajectory of length T, allowing $\pi_{master}$ at every time step (or on an interval) to either or continue using a sub-policy to act based on the advantage coordinate observed. The advantage for $\pi_{master}$, represented by $\mathbf{A}_t$, can be calculated using only the previous state-reward or it can be computed as a generalized estimate over the past $h$ time-steps as a rolling window. The following Eq. 5 describes the optimal policy for the master agent to choose action either staying the same policy or switching to another policy. While this form is similar to the generic policy in deep reinforcement learning, the only difference is conditioned on the MDP, which suggests the nominal or adversarial scenario.

$$\mathbf{A}_t = \big[A_{GAE,t-h}|\pi_{nom}, A_{GAE,t-h}|\pi_{adv}\big] \in \mathbb{R}^2 \qquad (4)$$

$$a_{master,t} = \pi_{*,t} = \underset{a}{\arg\max}\, \mathbb{E}_{s_t,\pi_i,m_i...}\Big[\sum_{t=0}^{T}\gamma^t r(s_t,a)|m_i\Big] \in \{0:\text{stay}, 1:\text{switch}\} \qquad (5)$$

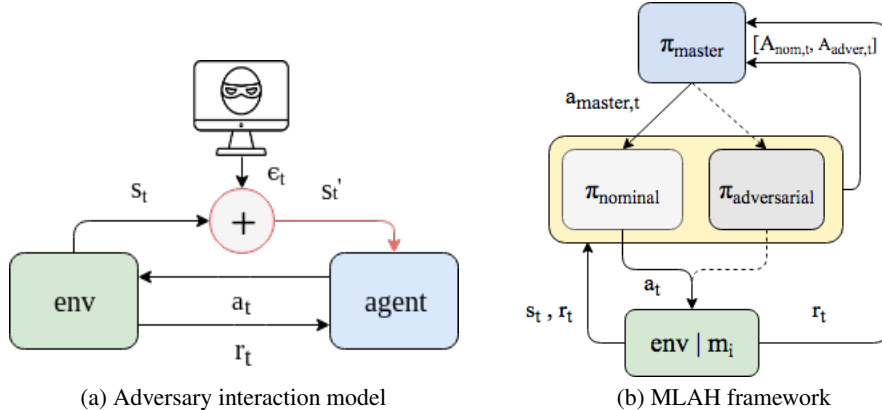

(a) Adversary interaction model          (b) MLAH framework

Figure 2: **a)** Illustration of the adversarial attack mechanism: corrupting the state observation, by injecting a perturbation $\epsilon$ before it reaches the agent, no perturbation in the reward signal. **b)** MLAH architecture: while similar to MLSH, key differences are: 1) master policy only observes the advantage of the sub-policy as a state and 2) only two sub-polices (nominal/adversarial) considered.

Observing the advantage over states and actions can be justified philosophically and has technical benefits when compared to other temporal state abstraction techniques that may be used to estimate the latent condition (RNN, LSTM). Although this mapping has potential to be noisy as the advantage can be trajectory dependent, it is static across the multiple sub-policies as opposed to a state-policy selection mapping which must be re-learned with every change in the latent condition.

**Advantage map as an effective metric to detect adversary:** To fool an RL agent into taking an alternative action, an adversary may use the policy network to compute a perturbation [18]. For attack mitigation, the RNN-based visual-foresight method [13] is practical, considering the predicted policy distance from the chosen policy. However, it was reported [13] that such a scheme can be fooled with a series of likely state perturbations. However in MLAH, even if the adversary could compute a series of likely states to fool the agent, the advantage would still be affected and the master agent may detect the attack. The adversary would have to consecutively fool the agent with a state that would be expected to give an equally bad reward. This constraint would make the perturbation especially hard or infeasible to compute. We do acknowledge however that this method is slightly delayed such that the agent has to experience an off-trajectory reward before it can detect the adversary presence and may also have to observe long attack periods before learning the advantage mapping.

## 4 Analysis of Bias Mitigation and Policy Improvement

Here we present analysis to show that the proposed MLAH framework reduces bias in the value function baseline under adversarial attacks. We then show how reducing bias is inherently beneficial for policy learning (improvement in expected reward lower bound compared to the state-of-the-art as

**Algorithm 1:** MLAH

---

**Input :** $\pi_{nom}$ and $\pi_{adv}$ sub-policies parameterized by $\theta_{nom}$ and $\theta_{adv}$; Master policy $\pi_{master}$ with parameter vector $\phi$.

1    Initialize $\theta_{nom}, \theta_{adv}, \phi$
2    **for** *pre-training iterations [optional]* **do**
3       |   Train $\pi_{nom}$ and $\theta_{nom}$ on only nominal experiences.
4    **end**
5    **for** *learning life-time* **do**
6       **for** *Time steps $t$ to $t + T$* **do**
7          |   Compute $\mathbf{A}_t$ over sub-policies (see eq. 4)
8          |   $\pi_{master}$ selects to switch or stay with sub-policy based on $\mathbf{A}_t$ observations to take action
9       **end**
10      Estimate all $A_{GAE}$ for $\pi_{nom}$, $\pi_{adv}$ over $T$
11      Estimate all $A_{GAE}$ for $\pi_{master}$ over $T$ with respect to $\mathbf{A}_t$ observations
12      Optimize $\theta_{nom}$ based on experiences collected from $\pi_{nom}$
13      Optimize $\theta_{adv}$ based on experiences collected from $\pi_{adv}$
14      Optimize $\phi$ based on all experiences with respect to $\mathbf{A}_t$ observations
15   **end**

---

presented in [11]) in the presence of adversaries. In order to estimate the expected value learned by a policy, we consider a first-order stochastic transition model (from nominal-0 to adversary-1 and vice versa) for the temporal profile of the attack as follows:

$$ P = \begin{bmatrix} p_{0|0} & p_{1|0} \\ p_{0|1} & p_{1|1} \end{bmatrix} = \begin{bmatrix} m & 1 - m \\ n & 1 - n \end{bmatrix} $$

This defines a Markov chain ($p_{b|a}$ denotes the probability transitioning from $a$ to $b$). Let the stationary distribution for this Markov chain be denoted by, $v = [p_0, p_1]$ that satisfies $v = vP$. Therefore,

$$ p_0 = \frac{n}{1 - m + n}, \qquad p_1 = \frac{1 - m}{1 - m + n} \tag{6} $$

which describes the long term expectation of visiting a nominal or adversarial state. As discussed in the preliminaries, trajectory experiences are handled with a distinct policy and value network when the adversarial attack is present. As the condition is perceived by the master agent, we can define two independent MDPs separately, i.e., one given a nominal state ($p_{\sim|0}$) and another given the perturbed state due to the adversary ($p_{\sim|1}$). With this setup, we present an assumption as follows:

**Assumption 1.** *Long term expectation of visiting a nominal state is higher than that of adversarial state, i.e., for the stochastic transition model P, $n < m$.*

Let $\mathbb{E}_{s \sim \mathcal{S}|0} V(s)$ be the expected discounted reward over states $\mathcal{S}$ given that the policy **only** sees nominal conditions ($m = 1$). Similarly, let $\mathbb{E}_{s \sim \mathcal{S}|1} V(s)$ be the expected discounted reward for the policy when it sees the adversarial states ($m = 0, n = 0$) alone. We simplify the notations as follows: $\mathbb{E}_{s \sim \mathcal{S}|0} V(s) = V_0$ and $\mathbb{E}_{s \sim \mathcal{S}|1} V(s) = V_1$ as two *value primitives*.

According to definition of the adversary (Definition 1), we have $V_1 < V_0$ as a successful adversarial attack leads to a sub-optimal return. We can now compare the expected discounted return for the *unconditioned* and *conditioned* learning scheme. Here, the unconditioned scheme refers to the learning scheme of a classical DRL agent with one policy. In this case, the expected discounted reward under adversarial attacks can be expressed as:

$$ \mathbb{E}_{unc, s \sim \mathcal{S}} V(s) = V_0 p_0 + V_1 p_1 = V_0 \frac{n}{1 - m + n} + V_1 \frac{1 - m}{1 - m + n} \tag{7} $$

On the other hand, the conditioned schemes refer to the two sub-policies (one given the nominal state and other given the adversarial state) based on the proposed MLAH framework. In this context, the expected discounted reward conditioned on the nominal state under adversarial attacks can be expressed as:

$$ \mathbb{E}_{con, s \sim \mathcal{S}|0} V(s) = V_0 p_{0|0} + V_1 p_{1|0} = V_0 m + V_1 (1 - m) \tag{8} $$

We now provide a lemma to compare the unconditioned and conditioned (given a nominal state) expected discounted rewards.

**Lemma 1.** *Let Assumption 1 hold. $\mathbb{E}_{unc, s \sim \mathcal{S}} V(s) < \mathbb{E}_{con, s \sim \mathcal{S}|0} V(s)$.*

See the proof in the Supplementary material.

We next discuss different lower bounds of the expected discounted rewards for the conditioned and unconditioned policies. We begin with defining the observed bias in the *state value* for both the conditioned and unconditioned policies by comparing the expected discounted reward to the original nominal value primitive $V_0$. Then, we have,

$$\delta_{con|0} = V_0 - \mathbb{E}_{con,s\sim\mathcal{S}|0}V(s) = (1-m)(V_0 - V_1), \quad \delta_{unc} = V_0 - \mathbb{E}_{unc,s\sim\mathcal{S}}V(s) = \frac{(1-m)(V_0 - V_1)}{1 - m + n}$$

With this setup, we present the following lemma.

**Lemma 2.** *Let Assumption 1 hold.* $\delta_{con|0} < \delta_{unc}$.

The proof is straightforward using Lemma 1 (see Supplementary material).

In this context, we express $V_0 = \mathbb{E}_{con,s\sim\mathcal{S}|0}V(s) + \delta_{con|0}$ and $V_0 = \mathbb{E}_{unc,s\sim\mathcal{S}}V(s) + \delta_{unc}$ in a general way as: $V(s) = \hat{V}(s) + \delta$, where $\delta$ is the observed bias in the state value. According to the definition of advantage function in Eq. 2, letting $\lambda = 0$, we have $A_\pi(s_t, a_t) = r_t + \gamma V(s_{t+1}) - V(s_t)$. Substituting $V(s) = \hat{V}(s) + \delta$ into the last equation yields

$$A_\pi(s_t, a_t) = r_t + \gamma\hat{V}(s_{t+1}) - \hat{V}(s_t) + \gamma\delta_{s,t+1} - \delta_{s,t} = \hat{A}_\pi(s_t, a_t) + \gamma\delta_{s,t+1} - \delta_{s,t} \quad (9)$$

where $\hat{A}_\pi(s_t, a_t)$ is the actual advantage function. While Lemma 2 shows that $\delta$ is reduced due to conditioning in our proposed framework, we note that the observed bias in the expected discounted reward can be different from that in the state value due to the complex and uncertain environment. Following the definition of the expected discounted reward in [11], recalling $V(s) = \hat{V}(s) + \delta$, the relationship between true and actual expected discounted reward is: $\mathcal{R}(\pi) = \mathbb{E}_{s\sim\pi}[\hat{V}_\pi(s_t, a_t) + \delta] = \hat{\mathcal{R}}(\pi) + \hat{\delta}$, where $\hat{\delta}$ is observed bias in the expected discounted reward. We denote the observed bias in the reward for the unconditioned and conditioned cases as: $\hat{\delta}_{unc}$ and $\hat{\delta}_{con|0}$. Let $\Delta\hat{\delta} = \hat{\delta}_{unc} - \hat{\delta}_{con|0}$ and $\Delta\delta = \delta_{unc} - \delta_{con|0}$. We are now ready to discuss the lower bounds of the expected discounted rewards for the conditioned and unconditioned schemes. Before that, based on [11], we introduce the maximum total variation divergence for any two different policies and use $\alpha$ to denote it for the rest of the analysis. We also first present one proposition to show the relationship between the actual expected discounted reward and its approximation. It is an extension of Theorem 1 in [11], which helps characterize the main claim in the paper.

**Proposition 1.** *Let Assumption 1 hold. Then the following inequality hold:*

$$\hat{\mathcal{R}}(\pi_{new}) \geq \hat{L}_{\pi_{old}}(\pi_{new}) - \frac{4\tilde{\epsilon}\gamma\alpha^2}{(1-\gamma)^2} \quad (10)$$

*where $\pi_{new}$ indicates the new policy, $\pi_{old}$ indicates the current policy, $\hat{L}_{\pi_{old}}(\pi_{new}) = L_{\pi_{old}}(\pi_{new}) + \delta - \hat{\delta}$, $L_{\pi_{old}}(\pi_{new})$ is the approximation of $\mathcal{R}(\pi_{new})$, i.e., $L_{\pi_{old}}(\pi_{new}) = \mathcal{R}(\pi_{old}) + \sum_s \rho_{\pi_{old}}(s)\sum_a \pi_{new}(a|s)A_{\pi_{old}}(s,a)$, $\rho$ is the discounted visitation frequencies as similarly defined in [11], $\tilde{\epsilon}$ satisfies the following relationship*

$$\tilde{\epsilon} = \begin{cases} max_{s,a}|\hat{A}_\pi(s,a)| + (\gamma - 1)\delta, & if \ \hat{A}_\pi(s,a) \geq (1-\gamma)\delta. \\ -max_{s,a}|\hat{A}_\pi(s,a)| + (1-\gamma)\delta, & if \ 0 < \hat{A}_\pi(s,a) < (1-\gamma)\delta. \\ max_{s,a}|\hat{A}_\pi(s,a)| + (1-\gamma)\delta, & if \ \hat{A}_\pi(s,a) \leq 0 \end{cases} \quad (11)$$

*Proof sketch*: Combining the proof of Lemmas 1, 2, and 3 in [11], we can immediately arrive at the similar form of conclusion as shown in Theorem 1 [11]. Then, we discuss the relationship between the actual advantage value $\hat{A}_\pi(s,a)$ and observed bias in the state value $(1-\gamma)\delta$ to complete the proof. Due to the limit of space, in this context we give the proof sketch and the complete proof can be found in the supplementary material and [26].

We then arrive at the following result to show that using the conditioned policy allows to achieve a higher lower bound of expected discounted reward.

**Proposition 2.** *If $\Delta\hat{\delta} < C\Delta V$, where $C \geq \frac{(m-n)(1-m)(4\gamma\alpha^2 + 1 - \gamma)}{(1-m+n)(1-\gamma)}$ and $\Delta V = V_0 - V_1$, then the conditioned policy has a higher lower bound of expected discounted reward compared to that of the unconditioned policy.*

*Proof sketch*: Based on Theorem 1 [11] and Proposition 1, we get the approximation of the actual expected discounted reward in both conditioned and unconditioned policies. Similarly we can obtain $\tilde{\epsilon}_{con|0}$ and $\tilde{\epsilon}_{unc}$. Due to the condition that $\hat{A}_\pi(s,a) \geq (1-\gamma)\delta$, we get the relationship between the approximation of the actual expected discounted reward and the observed bias in the expected discounted reward. Combining the condition that $\Delta\hat{\delta} < C\Delta V$, with some mathematical manipulation, the proof is completed. Due to the limit of space, we present the proof sketch and the complete proof can be found in the supplementary material and [26].

*Remark* 1. Proposition 1 suggests that under a certain condition, using the conditioned policy can improve the lower bound of the expected discounted return over the unconditioned policy. Intuitively, the condition demands the adversary to be sufficiently intelligent in order to have a large enough value for $\Delta V$.

# 5 Experimental Results

In order to justify the theoretical implications of bias reduction using a conditioned policy optimization, we implemented the proposed framework introduced in Section 3 with a selection of simple adversary models. Because the meta-learned framework has many moving parts and can be subject to instabilities, we first consider a case where the master agent is an oracle in determining the presence of an adversary. Then we consider the advantage-based adversary detection by the master agent.

## 5.1 Experimental Setup

For all experiments, we use the proximal clipped objective $L(\theta)^{CLIP+VF}$ from [17] instead of a constrained trust region optimization in accordance with recent results showing similar performance and ease of implementation. We use the same optimization for the master agent, although we acknowledge this may not be the best method for only two action choices (nominal or adversarial), we propose this to generalize to an arbitrary number of sub-polices. In every example, *training* denotes the agent acting with an $\epsilon$-greedy exploration policy with adversarial attacks. Simultaneously, we run an *evaluation* which executes a deterministic actions with the same policy, *without adversarial attacks*, hence obtain much higher return values. For the examples shown, we introduce the adversary on a fixed interval (e.g., 5000 with adversary, 10000 without). During that period, the adversary perturbs the state at *every* time step. For page limit constraints, PPO parameters used in experiments such as deep network size and actor-batches can be found in the supplementary material.

**Stochastic $l_\infty$-bounded Attacks:**

In this paper, for the purpose of experiment, we consider an attacker model that has the ability to perturb state information from the environment before it reaches the agent. Since gradient-based attacks for continuous action policies have not been thoroughly studied, we will focus on naive attacks which only sample the perturbation size and direction from a defined uniform distribution $\mathcal{U}(a,b)$ about the current state $\mathbf{s} = [s_0, s_1, \cdots, s_n]$. This results in an attack where $s_{i,\,adversary} = s_i + \mathcal{U}(a,b)$ where the perturbation is bounded by the $l_\infty$ norm so that $\forall s_i \in \mathbf{s} \quad \max_i |s_i - s_{i,\,adversary}| \leq \epsilon_{attack}$.

We find that this naive attack is effective enough to significantly decrease the return of a policy, although we do provide one example of an iterative gradient-based attack in the supplementary material Section 7.4. We specifically utilize white-noise attacks where $a = -b$ as well as bias attacks, where $a \neq b$ and $a < b$.

## 5.2 Adversarial Bias Reduction with MLAH

We begin by examining an RL environment where the master agent is asked to select the policy that corresponds to the current condition, i.e., nominal or adversarial. We acknowledge that this "policy" may not be the optimal master policy since a game may not be perfectly Markov. However, we find that this is sufficient to examine the policy improvement in some Openai Mujoco control environments [14].

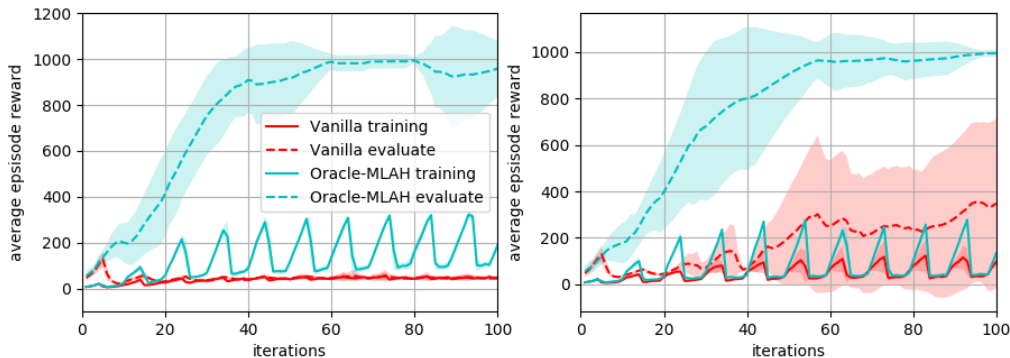

Figure 3: Results of Oracle-MLAH and Vanilla PPO applied to the InvertedPendulum-v2 game with repeatedly scheduled attacks for 5000 time steps and then off for 10000, displaying a $1\sigma$ bound. **Left:** Case study with an extreme bias attack spanning the entire state-space. Vanilla policy is unable to resolve the correct mapping due to large disturbances in the state information, while MLAH improves nearly monotonically. **Right:** Case study with a weaker bias attack, Vanilla agent still struggles.

Table 2: Performance evaluation of Oracle-MLAH

| | Normalized avg. training return | | Normalized avg. evaluation return | |
|---|---|---|---|---|
| $m/n$ | Vanilla | Oracle-MLAH | Vanilla | Oracle-MLAH |
| $1.0/-$ | $0.96 \pm 0.03$ | $0.96 \pm 0.03$ | $1.0$ | $1.0$ |
| $0.995/0.005$ | $0.238 \pm .082$ | $0.553 \pm 0.242$ | $0.471 \pm 0.051$ | $0.99 \pm 0.001$ |
| $0.95/0.05$ | $0.612 \pm .08$ | $0.677 \pm 0.149$ | $0.644 \pm 0.078$ | $0.99 \pm 0.001$ |
| $0.8/0.2$ | $0.613 \pm 0.043$ | $0.728 \pm 0.063$ | $0.539 \pm 0.023$ | $0.994 \pm 0.165$ |
| $0.5/0.5$ | $0.749 \pm 0.093$ | $0.764 \pm 0.078$ | $0.787 \pm 0.010$ | $0.948 \pm 0.086$ |

Comparison of the returns of Vanilla PPO and Oracle-MLAH under attacks over 40 policy optimization iterations with $1\sigma$ uncertainty bounds. The training return uses a stochastic policy for exploration and evaluation acts deterministically. The evaluation bias for the Oracle-MLAH remains substantially lower over all attack severity levels. Note when $m = n$, training returns are very similar as predicted by Eq. 8.

The returns shown in Table 2 and Figure 3 for long and intermittent bias attacks (large m and small n) clearly demonstrate the benefit of using distinct policies for nominal and adversarial states respectively. According to eq. 8, this attack condition produces the largest difference in bias between conditioned and unconditioned policies. As a policy can only solve for one state-action mapping and there are clearly two separate MDP state-reward distributions existing across time, a singly policy has no choice, but to optimize over the mean of these two distributions. Often times this results in not developing a useful policy for either condition as shown in figure 3. Enabling the use of multiple polices in this intermittent attack case allows the agent to optimize for both mappings, even learning to mitigate the reduced return during the adversarial attack. More simulation results using Open Gym environments such as *MountainCarContinuous-v0* and *Hopper-v2* [14] are included in the supplementary material.

It can be seen in table 2 that as the switching expectations between nominal and adversarial states rise, the unconditioned (Vanilla) policy actually performs increasingly well, but still less than that of the conditioned (MLAH) policy. This is perhaps because the switching is quick enough to map the scenario to one state-reward distribution, which is favorable for a single policy agent.

As anticipated by the analysis, when $m = n$, the training performances of both policies approach a similar value, however the conditioned MLAH agent was able to maintain a nearly unbiased evaluation return. This may be an artifact of the environment or adversary, which is relatively simple and unintelligent. Over longer attack periods, it may be unrealistic to expect the return to behave according to the stationary distribution expectation because the average resolves on a longer time scale than policy optimization.

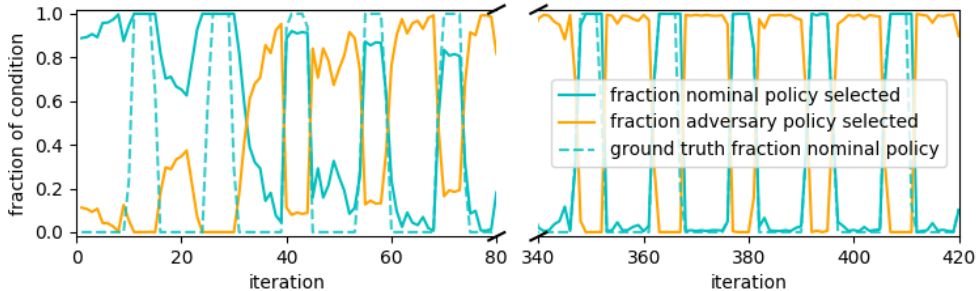

Figure 4: Master agent's performance in learning from two random policies to decide which to employ to maximize the reward of *InvertedPendulum-v2* with bounded 5000 on, 10000 off bias attacks. The master agent is not given any information on which states are perturbed by the adversary. After initial learning, the policy choices clearly diverge during the attack intervals with few exceptions.

Next we put our master agent to the test, using the relative advantage coordinate mappings. This formulation is a novel alternative to previous meta-learned hierarchies which are non-stationary and need to be reset over time [25]. The relative advantage mapping is stationary across multiple MDPs under certain conditions. In order for the master agent to arrive at correct advantage-policy mapping, the policies themselves must also optimize to produce better advantage estimates in this expectation maximization (EM) type algorithm. This makes it challenging to produce a stable learning sequence of polices and advantage mappings. However, this mapping can be learned from "nothing" if an adversary creates a strong enough presence by altering the state-reward mapping (by Definition 1). This optimization process is explained in more depth in the supplementary material. Depending on whether the nominal policy is pre-trained and the effectiveness of the adversary, the meta agent can reliably use each policy during the respective conditions. As seen in Figure 4, an adversary is introduced in an intermittent manner and the master agent has two random sub-polices at its disposal.

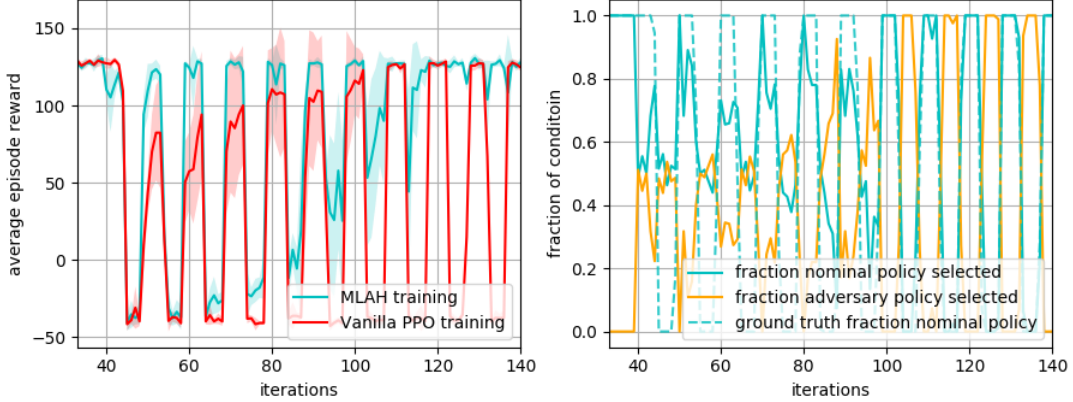

Figure 5: Shown is MLAH simultaneously learning to switch policies and a mitigation strategy on a small $11 \times 11$ grid world. The adversary simply gives the agent a deterministic mirrored column observation about the center of the grid, making it so that the optimal policy is different for every state given there is an attack. The attacks are applied intermittently on intervals of $5000$ actions, showing $1 - \sigma$ variance.

The agent optimizes to use one policy for the nominal and the other for the adversarial conditions to optimize its reward. The policy-selection results in Figure 4 may resemble a Bayesian non-parametric latent state estimator [27]. However, being entirely in the context of RL, MLAH is unique and uses the advantage observation and a meta-learning objective to form a belief over the latent conditions.

### 5.3 Countering Deterministic Attack Strategies

Given a deterministic attack strategy, it is likely that there is a learnable counter policy that performs optimally once the attack is detected. To clearly demonstrate MLAH we create a GridWorld environment and an attack which simply reflects the agents column position about the center of the grid. This attack is interesting to us because it is completely deterministic and, for the single policy Vanilla PPO, it is impossible to be optimal in both nominal and adversarial conditions, requiring an additional mirrored $\mathcal{S} \to \mathcal{A}$ map. In this experiment we let the policy train in the nominal environment with no attack for 40 iterations ($\sim 40000$ actions). Once attacks are introduced, it only takes MLAH about 50000 actions to, not only solve the meta task of switching policies, but also learn the adversary mitigation policy from a random initial policy. The success of both of these task strongly depend on each other, that is a noisy value function will result in poor policy switching and vice versa. This is quite evident by the variation in reward when MLAH is learning the attack strategy and switching policy simultaneously in 5. There may be further improvements to address the stability of this algorithm for more complicated tasks.

## 6   Conclusions

We have discussed a new MLAH framework for handling adversarial attacks in an online manner specifically in the context of RL. This framework is attack-model agnostic and presents a general way of examining adversarial attacks in the temporal domain. Analyzing the hierarchical policy MLAH in this way, we can show that under certain conditions, the return lower-bound is improved when compared to a single policy agent. In future research, we aim to improve the stability of MLAH by optimizing the master agent function, perhaps using a more simple method to regress the advantage space. We will also attempt to extend MLAH to a more general framework for decision problems with multiple time-varying objectives.

**Acknowledgements**

This work has been supported in part by the U.S. AFOSR under the YIP grant FA9550-17-1- 0220. Any opinions, findings and conclusions or recommendations expressed in this publication are those of the authors and do not necessarily reflect the views of the sponsoring agencies.

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
