[Supplementary Material · supplementary.pdf]

# 7 Supplementary Materials

## 7.1 Additional Analysis

This section presents the analysis for all of lemmas and propositions and additional analysis.
**Transition Mechanism of Adversary MDP**:

Figure 6: Assumed mechanism (only for analysis purpose) for nominal to adversary state transitions: $p_{0|0}$ signifies the probability that a nominal state transits to another nominal state; $p_{1|0}$ signifies the probability that a nominal state transits to an adversarial state; $p_{0|1}$ signifies the probability that an adversarial state transits to a nominal state; $p_{1|1}$ signifies the probability that an adversarial state transits to another adversarial state

The rest of the analysis in this section is based on the above transition mechanism.
**Proof of Lemma 1**:

*Proof.* Based on the definitions of $\mathbb{E}_{unc,s\sim\mathcal{S}}V(s)$ and $\mathbb{E}_{con,s\sim\mathcal{S}|0}V(s)$, we have

$$\mathbb{E}_{unc,s\sim\mathcal{S}}V(s) - \mathbb{E}_{con,s\sim\mathcal{S}|0}V(s) = V_0\frac{n}{1-m+n} + V_1\frac{1-m}{1-m+n} - V_0 m - V_1(1-m)$$

With some mathematical manipulation, we have

$$\mathbb{E}_{unc,s\sim\mathcal{S}}V(s) - \mathbb{E}_{con,s\sim\mathcal{S}|0}V(s) = \frac{(V_0-V_1)(n-m)(1-m)}{1-m+n} \tag{12}$$

As $V_1 < V_0$ and $n < m$, then we get the desired results. $\qquad\square$

**Proof of Lemma 2**:

*Proof.* As $V_1 < V_0$, then $V_0 - V_1 > 0$. Based on the definitions of $\delta_{con|0}$ and $\delta_{unc}$, and Lemma 1, the desired result is immediately obtained. $\qquad\square$

The following analysis is for establishing the relationship between the true and actual expected discounted rewards.
For completeness, we rewrite or redefine some definitions here to characterize the analysis. We denote by $\hat{V}(s)$ the actual state value of the learned policy (i.e., the conditioned or unconditioned). Define the relationship between the true state value and actual state value as:

$$V(s) = \hat{V}(s) + \delta$$

which can be adaptive to the unconditioned or conditioned policy by substituting different bias. $\delta$ is the observed bias in the state value. We also denote by $\pi$ and $\tilde{\pi}$ the current policy and the new policy. According to the definition of advantage function in Eq. 2, letting $\lambda = 0$, we have

$$A_\pi(s_t, a_t) = r_t + \gamma V(s_{t+1}) - V(s_t)$$

Substituting $V(s) = \hat{V}(s) + \delta$ into the last equation yields

$$A_\pi(s_t, a_t) = r_t + \gamma\hat{V}(s_{t+1}) - \hat{V}(s_t) + \gamma\delta_{s,t+1} - \delta_{s,t} = \hat{A}_\pi(s_t, a_t) + \delta_{s,t+1}(\gamma-1) \tag{13}$$

where $\hat{A}_\pi(s_t, a_t)$ is the actual advantage function under a learned policy. Based on the definition of expected discounted reward in [11], we have

$$\mathcal{R}(\pi) = \mathbb{E}_{s\sim\pi}\left[V_\pi(s_t, a_t)\right] \tag{14}$$

which results in the relationship between the true and actual expected discounted rewards as follows

$$\mathcal{R}(\pi) = \mathbb{E}_{s\sim\pi}\left[\hat{V}_\pi(s_t, a_t) + \delta\right] = \hat{\mathcal{R}}(\pi) + \hat{\delta} \tag{15}$$

where $\hat{\delta}$ is the observed bias in the expected discounted reward. It is immediately obtained that corresponding to different learned policies, $\hat{\delta}$ is not the same. In this context, we define $\hat{\delta}_{unc}$ as the observed bias in the expected discounted reward caused by the unconditioned policy and $\hat{\delta}_{con|0}$ as the observed bias in the expected discounted reward caused by the conditioned policy. Now we analyze the expected discounted reward of the new policy $\tilde{\pi}$ in terms over the current policy $\pi$ in order to know the difference between different policies during the learning process. Following [11], we define the expected discounted reward of $\tilde{\pi}$ as follows

$$\mathcal{R}(\tilde{\pi}) = \mathcal{R}(\pi) + \mathbb{E}_{s,a\sim\hat{\pi}}\left[\sum_{t=0}^{T}\gamma^{t}A_{\pi}(s_t, a_t)\right] \tag{16}$$

Hence, combining Eq. 13 and Eq. 16 we obtain the expected discounted reward of the new policy $\tilde{\pi}$ with respect to the expected discounted reward of the current policy $\pi$, the actual advantage and the observed bias in the state value.

$$\mathcal{R}(\tilde{\pi}) = \mathcal{R}(\pi) + \mathbb{E}_{s,a\sim\tilde{\pi}}\left[\sum_{t=0}^{T}\gamma^{t}\hat{A}_{\pi}(s_t, a_t)\right] + \mathbb{E}_{s,a\sim\tilde{\pi}}\left[\sum_{t=0}^{T}\gamma^{t}(\gamma\delta_{s,t+1} - \delta_{s,t})\right] \tag{17}$$

As we use the same neural networks to estimate the actual state values, we assume that in Eq. 17 the expectation of bias $\delta_{s,t}$ given the state $s$ can be treated equally as constant, represented by $\delta$ for convenience of analysis. Therefore, by substituting Eq. 15 the last equality becomes as follows

$$\mathcal{R}(\tilde{\pi}) = \hat{\mathcal{R}}(\pi) + \hat{\delta} + \mathbb{E}_{s,a\sim\tilde{\pi}}\left[\sum_{t=0}^{T}\gamma^{t}\hat{A}_{\pi}(s_t, a_t)\right] - \delta = \hat{\mathcal{R}}(\tilde{\pi}) + \hat{\delta} - \delta \tag{18}$$

which shows the true expected discounted reward of the policy $\tilde{\pi}$ with respect to its actual expected discounted reward $\hat{\mathcal{R}}(\tilde{\pi})$, the observed bias in the expected discounted reward, $\hat{\delta}$, and the observed bias in the state value, $\delta$.

For the rest of analysis, we follow the similar analysis procedure presented [11] and for convenience we denote by $\pi_{old}$ the current policy $\pi$ and by $\pi_{new}$ the new policy $\tilde{\pi}$. Following [11], we first rewrite Eq. 16 as the following equation

$$\mathcal{R}(\pi_{new}) = \mathcal{R}(\pi_{old}) + \sum_{s}\sum_{t=0}^{T}\gamma^{t}\mathcal{P}(s_t = s|\pi_{new})\sum_{a}\pi_{new}(a|s)A_{\pi_{old}}(s, a)$$
$$= \mathcal{R}(\pi_{old}) + \sum_{s}\rho_{\pi_{new}}(s)\sum_{a}\pi_{new}(a|s)A_{\pi_{old}}(s, a) \tag{19}$$

where $\rho_{\pi_{new}}$ is the discounted visitation frequencies as similarly defined in [11]. Then we define an approximation of $\mathcal{R}(\pi_{new})$ as

$$L_{\pi_{old}}(\pi_{new}) = \mathcal{R}(\pi_{old}) + \sum_{s}\rho_{\pi_{old}}(s)\sum_{a}\pi_{new}(a|s)A_{\pi_{old}}(s, a) \tag{20}$$

due to the complex dependence of $\rho_{\pi_{new}}$ on $\pi_{new}$. Similarly, according to Eq. 18 we have

$$L_{\pi_{old}}(\pi_{new}) = \hat{L}_{\pi_{old}}(\pi_{new}) + \hat{\delta} - \delta \tag{21}$$

For completeness, we state the main theorem from [11] to guarantee the monotonic improvement. Before that, we need to define the total variation divergence for two different discrete probability distributions $q, o$, i.e., $D_{TV}(q||o) = \frac{1}{2}\sum_{i}|q_i - o_i|$, based on which, we define $D_{TV}^{max}(\pi_{old}, \pi_{new}) = max_s D_{TV}(\pi_{old}(\cdot|s)||\pi_{new}(\cdot|s))$. Following [11], we state the main theorem from [11] to guarantee the monotonic improvement.

**Theorem 1.** *(Theorem 1 in [11]) Let $\alpha = D_{TV}^{max}(\pi_{old}, \pi_{new})$. Then the following bound holds:*

$$\mathcal{R}(\pi_{new}) \geq L_{\pi_{old}}(\pi_{new}) - \frac{4\epsilon\gamma\alpha^2}{(1-\gamma)^2} \tag{22}$$

*where $\epsilon = max_{s,a}|A_{\pi}(s, a)|$.*

With this, we arrive at the following proposition to demonstrate the relationship between the actual expected discounted reward and its approximation.

**Proposition 1** Let $\alpha = D_{TV}^{max}(\pi_{old}, \pi_{new})$. Then the following inequality hold:

$$\hat{\mathcal{R}}(\pi_{new}) \geq \hat{L}_{\pi_{old}}(\pi_{new}) - \frac{4\tilde{\epsilon}\gamma\alpha^2}{(1-\gamma)^2} \tag{23}$$

where $\tilde{\epsilon}$ satisfies the following relationship

$$\tilde{\epsilon} = \begin{cases} max_{s,a}|\hat{A}_\pi(s,a)| + (\gamma-1)\delta, & \text{if } \hat{A}_\pi(s,a) \geq (1-\gamma)\delta. \\ -max_{s,a}|\hat{A}_\pi(s,a)| + (1-\gamma)\delta, & \text{if } 0 < \hat{A}_\pi(s,a) < (1-\gamma)\delta. \\ max_{s,a}|\hat{A}_\pi(s,a)| + (1-\gamma)\delta, & \text{if } \hat{A}_\pi(s,a) \leq 0 \end{cases} \tag{24}$$

*Proof.* Combining Eq. 21 with the proof of Lemmas 1, 2, and 3 in [11], we can arrive at the similar form of conclusion as shown in Theorem 1. The difference between the conclusion in Theorem 1 and Proposition 1 is when we consider the actual expected discounted reward, the $\tilde{\epsilon}$ value is different from the $\epsilon$ value in Eq. 22. We next discuss the new value for $\tilde{\epsilon}$. As the advantage function has the following relationship

$$A_\pi(s_t, a_t) = \hat{A}_\pi(s_t, a_t) + \delta(\gamma-1)$$

Then, $\tilde{\epsilon} = max_{s,a}|\hat{A}_\pi(s,a) + (\gamma-1)\delta|$. Since $\delta > 0$ and $\gamma - 1 < 0$, we need to discuss the sign of $\hat{A}_\pi(s,a) + (\gamma-1)\delta$. Three cases are discussed as below:

1. When $\hat{A}_\pi(s,a) + (\gamma-1)\delta \geq 0$ such that $\hat{A}_\pi(s,a) \geq (1-\gamma)\delta$, $\tilde{\epsilon} = max_{s,a}|\hat{A}_\pi(s,a)| + (\gamma-1)\delta$,

2. When $\hat{A}_\pi(s,a) + (\gamma-1)\delta \leq 0$ and if $0 < \hat{A}_\pi(s,a) < (1-\gamma)\delta$, $\tilde{\epsilon} = -max_{s,a}|\hat{A}_\pi(s,a)| + (1-\gamma)\delta$,

3. When $\hat{A}_\pi(s,a) + (\gamma-1)\delta \leq 0$ and if $\hat{A}_\pi(s,a) \leq 0$, $\tilde{\epsilon} = max_{s,a}|\hat{A}_\pi(s,a)| + (1-\gamma)\delta$,

which completes the proof. $\square$

*Remark* 2. The condition $\hat{A}_\pi(s,a) \geq (1-\gamma)\delta$ above may seem constrictive, but it can hold. If we consider that in order to achieve a positive advantage, the value function must be biased to underestimate the reward at the beginning. Therefore, the value function bias itself needs to be biased by *at least* $\delta(1-\gamma)$ at the beginning. Hence, for any $\delta > 0$, we have $\delta(1-\gamma) < \delta$, which is always true as $0 < \gamma < 1$. One can arrive at the same result for $\delta < 0$ when $\hat{A}_\pi(s,a) \leq (1-\gamma)\delta$.
Now we will show the Proposition 1 with the condition $\hat{A}_\pi(s,a) \geq (1-\gamma)\delta$.
**Proof of Proposition 2**:

*Proof.* For assessing the new lower bound, we have exactly accounted for the bias in both conditioned and unconditioned policies. Therefore, according to Theorem 1, Eq. 21, and Eq. 23, we have

$$\begin{aligned} L_{\pi_{old}}(\pi_{new}) - \frac{4\epsilon\gamma\alpha^2}{(1-\gamma)^2} &= \left(\hat{L}_{\pi_{old}}(\pi_{new})\right)_{con|0} + \hat{\delta}_{con|0} - \delta_{con|0} - \frac{4\tilde{\epsilon}_{con|0}\gamma\alpha^2}{(1-\gamma)^2} \\ &= \left(\hat{L}_{\pi_{old}}(\pi_{new})\right)_{unc} + \hat{\delta}_{unc} - \delta_{unc} - \frac{4\tilde{\epsilon}_{unc}\gamma\alpha^2}{(1-\gamma)^2} \end{aligned} \tag{25}$$

The $\left(\hat{L}_{\pi_{old}}(\pi_{new})\right)_{con|0}$ and $\left(\hat{L}_{\pi_{old}}(\pi_{new})\right)_{unc}$ signify the approximation of $\hat{\mathcal{R}}(\pi_{new})$ in both conditioned and unconditioned policies, respectively. Similarly, $\tilde{\epsilon}_{con|0}$ and $\tilde{\epsilon}_{unc}$ indicate the different upper bounds corresponding to the conditioned and unconditioned policies, respectively. Let $\hat{\epsilon} = max_{s,a}|\hat{A}_\pi(s,a)|$ such that we have $\hat{\epsilon}_{con|0}$ and $\hat{\epsilon}_{unc}$ for the conditioned and unconditioned policies. Due to the condition that $\hat{A}_\pi(s,a) \geq (1-\gamma)\delta$, based on Proposition 1 we have

$$\tilde{\epsilon}_{unc} = \left(max_{s,a}|\hat{A}_\pi(s,a)|\right)_{unc} + (\gamma-1)\delta_{unc} = \hat{\epsilon}_{unc} + (\gamma-1)\delta_{unc} \tag{26}$$

and

$$\tilde{\epsilon}_{con|0} = \left(max_{s,a}|\hat{A}_\pi(s,a)|\right)_{con|0} + (\gamma-1)\delta_{con|0} = \hat{\epsilon}_{con|0} + (\gamma-1)\delta_{con|0} \tag{27}$$

Hence, substituting Eq. 26 and Eq. 27 into Eq. 25, we have

$$
\left(\hat{L}_{\pi_{old}}(\pi_{new})\right)_{con|0} + \hat{\delta}_{con|0} - \delta_{con|0} - \frac{4\hat{\epsilon}_{con|0}\gamma\alpha^2}{(1-\gamma)^2} + \frac{4\delta_{con|0}\gamma\alpha^2}{1-\gamma}
$$
$$
= \left(\hat{L}_{\pi_{old}}(\pi_{new})\right)_{unc} + \hat{\delta}_{unc} - \delta_{unc} - \frac{4\hat{\epsilon}_{unc}\gamma\alpha^2}{(1-\gamma)^2} + \frac{4\delta_{unc}\gamma\alpha^2}{1-\gamma}
\tag{28}
$$

By the condition that $\Delta\hat{\delta} < C\Delta V$ and $\Delta\hat{\delta} = \hat{\delta}_{unc} - \hat{\delta}_{con|0}$, we have

$$
\begin{aligned}
\hat{\delta}_{unc} - \hat{\delta}_{con|0} &< \frac{(m-n)(1-m)(4\gamma\alpha^2+1-\gamma)}{(1-m+n)(1-\gamma)}\Delta V \\
&= \Delta V\frac{(m-n)(1-m)}{1-m+n}\frac{4\gamma\alpha^2+1-\gamma}{1-\gamma} \\
&= \Delta V\frac{m-n+mn-m^2+1-m+m-1}{1-m+n}\frac{4\gamma\alpha^2+1-\gamma}{1-\gamma} \\
&= \Delta V\frac{1-m-(1-m+n-m+m^2-mn)}{1-m+n}\frac{4\gamma\alpha^2+1-\gamma}{1-\gamma} \\
&= \Delta V\frac{1-m-(1-m)(1-m+n)}{1-m+n}\frac{4\gamma\alpha^2+1-\gamma}{1-\gamma} \\
&= \left(\frac{(1-m)\Delta V}{1-m+n} - (1-m)\Delta V\right)\left(\frac{4\gamma\alpha^2}{1-\gamma}+1\right)
\end{aligned}
\tag{29}
$$

According to the definition of bias for the expected discounted reward, we have

$$
\begin{aligned}
\hat{\delta}_{unc} - \hat{\delta}_{con|0} &< (\delta_{unc} - \delta_{con|0})\left(\frac{4\gamma\alpha^2}{1-\gamma}+1\right) \\
&= \frac{4\gamma\delta_{unc}\alpha^2}{1-\gamma} - \frac{4\gamma\delta_{con|0}\alpha^2}{1-\gamma} + \delta_{unc} - \delta_{con|0}
\end{aligned}
\tag{30}
$$

The last inequality yields the following relationship:

$$
\frac{4\gamma\delta_{con|0}\alpha^2}{1-\gamma} + \delta_{con|0} - \hat{\delta}_{con|0} < \frac{4\gamma\delta_{unc}\alpha^2}{1-\gamma} + \delta_{unc} - \hat{\delta}_{unc}
\tag{31}
$$

which results in the next inequality, combined with Eq. 28

$$
\left(\hat{L}_{\pi_{old}}(\pi_{new})\right)_{con|0} - \frac{4\hat{\epsilon}_{con|0}\gamma\alpha^2}{(1-\gamma)^2} > \left(\hat{L}_{\pi_{old}}(\pi_{new})\right)_{unc} - \frac{4\hat{\epsilon}_{unc}\gamma\alpha^2}{(1-\gamma)^2}
\tag{32}
$$

which suggests that by the conditioned policy, the lower bound of expected discounted reward is higher. It completes the proof. $\square$

## 7.2 Meta Optimization of the Advantage Space

To better explain the use of the advantage coordinate space, we provide some additional illustrations of the interesting optimization process at play. For visualization purposes, in figure 7 we simulated a value surface with injected noise for a game in which there are two goal positions on a 2D plane, one at $[-1, 0]$ and the other at $[1, 0]$. At any moment the goal may be at only one of these positions. When we create two polices to learn each distinct goal and value surface, we start from nothing, and the advantage space extremely noisy. The master agent will try its best to select sub-policies given this mapping and incrementally, each policy will become slightly better, meaning there is less bias and variance in the value function predictor. This will then allow the master agent to select policies with even greater accuracy, in result, improving the two value function accuracy more. One can see from this iterative process that it can hopefully achieve both an accurate master, and high-performing distinct polices simultaneously. This is an interesting way to perform an EM style optimization because it is *only* defined by a reward signal. All other optimization steps can be derived from that single scalar signal.

Figure 7: Depicted is the visualized meta optimization process for a multi-objective game. In each frame there are two surface, each representing the belief of the state-value map. As the map becomes more accurate, the advantage space becomes easier to regress, which improves value accuracy and so on. One can draw parallels to some sort of expectation maximization style algorithm.

## 7.3 Additional Results

In this section, we provide some additional experiments, results and illustrations that may help the reader better understand the implications of the paper. We have tested the oracle-MLAH based protocol on several Gym environments and of these we show *MountainCarContinuous-v0* Figure 8 and *Hopper-v2* Figure 10 case studies with white noise attacks and discuss some interesting observations made in each. All experiment are once again using the PPO clipped objective function with value prediction bonus.

### 7.3.1 MountainCarContinuous-v0

This experiment using the *MountainCarContinuous-v0* environment was particularly insightful due to the behavior of the bias. The Vanilla policy was able to achieve considerable reward and the difference between it's nominal and adversarial peaks was small. It can be noted that out of the average maximum nominal reward of $\approx 95.0$ and a minimum with adversaries $\approx 5.0$, the expected biased return should have been $30.0$ according to $\Delta V$ from eq. 8. This is approximately the observed average return for the Vanilla policy.

Figure 8: Here we show an adversary that is implements strong stochastic white noise on the MountainCarContinuous-v0 environment. The baseline for the adversarial transitions happens to be approximately $5.0$ which makes our $\Delta V$ about $90.0$ reward points. According to the approximate $m$ and $n$ for this experiment, the bias should be about $60.0$ reward points ($30.0$ return), which is approximately the mean return for the Vanilla policy.

### 7.3.2 CartPole-v1 Gradient Attack

We introduce a gradient attack similar to [15] designed for DDPG policies, but slightly modified to work on our value function based policies. The attack is iterative based on the nominal value network parameters. We take the state as input and take step size $\alpha$ in the opposite direction of the gradient of the value network in respect to that state input. Although this attack is rather naive, the attack strategy will change over time via the value network parameters as the nominal policy tries to adapt. We assume that the adversary is not aware of the adversary policy used in MLAH. Because MLAH will separate all information related to mitigating the attack into the adversary policy, the nominal policy is not biased and a separate mapping can be solved to counter the gradient attack. Note that in this example we allow an initial pre-training period in the nominal condition, so that the there is a consistent basis for the gradient attack.

$$x_{k+1} = x_k - \alpha \nabla_x V(x_k) \tag{33}$$

Figure 9: A gradient attack based on the nominal value function network is implemented and is introduced on an on/off interval of 5000 action steps after an initial pre-training period. We observe the MLAH algorithm with perfect switching can rather quickly learn a counter adversary policy which leaves the nominal policy unbiased. The Vanilla PPO algorithm will struggle to maintain the nominal baseline and adapt to the changing attack.

### 7.3.3 Hopper-v2

The *Hopper-v2* environment behaved similar to others during the attacks, except that the average performance during the attack appeared to be lower for the MLAH oracle, however MLAH oracle remained less biased in the nominal case. This is a curious observation that tells us that MLAH may not always mitigate the attack as well as a single Vanilla policy (in this case PPO).

### 7.4 Parameter Settings

Table 3: Sub Policy Hyperparameters

| Parameter | Value |
|---|---|
| Horizon (T) | 1024-2048 |
| Discount rate ($\gamma$) | 0.99 |
| GAE discount rate ($\lambda$) | .98 |
| Clipping parameter($\epsilon$) | 0.2 |
| Surrogate opt. Batch Size | 64 |
| Surrogate opt. Epochs | 10 |
| Adam stepsize | $3e\text{-}4$ |
| Hidden layers (Fully Connected) | 2 |
| Hidden nodes (tanh) | 32 |

Table 4: Master Hyperparameters

| Parameter | Value |
|---|---|
| Horizon (T) | 1024-2048 |
| Discount rate ($\gamma$) | 0.99 |
| GAE discount rate ($\lambda$) | .98 |
| Clipping parameter($\epsilon$) | 0.2 |
| Surrogate opt. batch size | 64 |
| Surrogate opt. epochs | 10 |
| Adam stepsize | 0.1 |
| Hidden layers (Fully Connected) | 2 |
| Hidden nodes (tanh) | 16 |

Figure 10: Here we show an adversary that is implements strong stochastic white noise on the Hopper-v2 environment. This environment-adversary pair is particularly interesting because it shows that the unconditioned policy actually learned to handle the adversary more effectively than the conditioned MLAH. However, it obviously suffers in the nominal condition, while MLAH receives significantly higher returns.