[Reviews · NeurIPS 2018]

Reviewer 1



It was difficult for me to understand the motivation. Even after thoroughly reading this paper, I could not imagine a simple real world example of this framework. It may be due to the lack of my background knowledge, but I expect a description of such a real world situation in the introduction. Partially related, I could not imagine a situation that an adversary can change an agent's state, not just an observation of the agent. It is written in the paper (Definition 1) that only the observation of the agent is affected, but in the experiments it seems that the actual state is corrupted by a noise. Please make these points clearer. [Rebuttal] My only concern was the lack of the description of the motivative real world examples, and the authors promised to address them in the introduction. The authors responses to the other reviewers are also reasonable. In the end, I expect that the final version of this paper will be rather improved.

Reviewer 2



Summary: This work considers the problem of learning a robust policy in a scenario where state inputs to the policy are subject to intermittent periods of adversarial attack. The authors propose a meta-learning based approach, whereby separate sub-policies are learned for the nominal and adversarial conditions, and at each time step a master policy selects one of these sub-policies to execute on the basis of their respective advantage estimates for the current observed state. Qualitative assessment: The idea of using advantage estimates to detect adversarial attacks is quite appealing, since by definition an adversarial attack should decrease the advantage of the policy regardless of the attack method used. However, a disadvantage is that the attack can only be detected after a delay, since the agent must first experience an unexpectedly low reward. This seems like it would be especially problematic in domains with long time horizons and sparse rewards, where the consequences of selecting a suboptimal action may not become apparent until much later on. The authors acknowledge this limitation, although I think it deserves more than a sentence of discussion. I’d be interested to see how well their method performs on tasks where the reward is sparse. The adversarial attacks simply consisted of additive uniform noise, which seem rather crude. The authors state that gradient-based attacks are not well-studied in the context of continuous action policies. I think this is fair, although at least one gradient-based attack against DDPG has been described in the literature (Pattanaik et al., 2017). One of the purported advantages of the method described in this paper is that it is agnostic to the attack model, so it would be good if the authors could demonstrate this by showing that it is also effective at mitigating a different type of adversarial attack. Figure 4 shows a single example where the MLAH master agent has learned to distinguish between the nominal and adversarial conditions. However the more salient thing to show is the overall performance of the non-oracle MLAH agent. In particular I’d like to see learning curves like those in Figure 3 plotted on the same axes as oracle-MLAH so that we can see the performance impact of learning the master policy rather than using an oracle to select the sub-policy. I'd also like to get a sense of what coping strategies are learned by π_adv. For example, can π_adv learn to correct for biases introduced by adversarial perturbations, or does it simply learn an open-loop policy that ignores the input states? On a similar note it would be good to see some videos showing the behavior of π_nom and π_adv under nominal and adversarial conditions. Clarity: The main ideas in the paper are presented quite clearly, although there are a few details that I think ought to be included: - Table 2 and Figure 3 show 1σ uncertainty bounds, but the authors don’t state how many random seeds were used per experimental condition. - The magnitudes of the adversarial perturbations aren’t stated explicitly. The caption of Figure 3 refers to an “extreme bias attack spanning the entire state space” and a “weaker bias attack”, but it’s unclear what exactly this mean in terms of the scale of the observations. - The text mentions that the PPO parameters are given in the supplementary material, but this information seems to be missing. Minor comments: - Line 55: “We compare our results with the state-of-the-art PPO that is sufficiently robust to uncertainties to understand the gain from multi-policy mapping.” - this sentence doesn't make sense to me. - Line 115: apposed → opposed - Line 256: executes deterministic actions - Line 265: optimize its attack - Line 281: singly → single

Reviewer 3



This submission introduces a novel framework for handling adversarial attacks on the observation space used by RL value estimation techniques. The new method is particularly interesting because it claims to address a few issues with current methods: the proposed method is attack-model agnostic, online, and adaptive. Although I find the work very interesting and I consider the problems that the paper is studying very important, I am not very familiar with most of the work that the authors have cited, in part because this work builds on many ArXiv pre-prints that I was not aware of. I find this paper to be very hard to read for someone who is not familiar with the latest results in addressing adversarial attacks on Deep RL. The authors chose some weird notational standards, the algorithmic framework is carelessly presented, and the main theoretical result is barely interpreted or compared to previous work. Although I appreciate the empirical results and the claimed improvements, I don't think this paper is very accessible. Please see a list of specific comments. L25, 55: is there a reason why some citations are listed separately? I.e. why [9], [10] and not [9, 10]? L28-31: Can you add some references to support the first statement in the Introduction? Particularly, the phrase "even small perturbations in state inputs can result in significant loss ..." seems like something that should have been quantified or empirically demonstrated already. L34: "the the" -> "the" L43: What does it mean to "optimize the observation space" ? Please fix this ambiguity. Table 1: It seems odd to add the column on "Mitigation", as all the listed methods support it. Also, what is the difference between a "robust method" and "a method supporting mitigation"? L49: Please define the TRPO acronym (e.g. on L27) L55, L85, L174: You should avoid using the term "state-of-the-art" in isolation. Both the context (i.e. environment type, assumptions) and the comparison metric should always be clear whenever a method is considered state-of-the-art. L65: "in [20] in" -> "in [20] for" Section 2: The MDP notation is very sloppy here. E.g. * It is unclear whether you assume deterministic policies, * Are we dealing with episodic tasks? Is T fixed, probabilistic? * The convention is most of the literature is to use "E_{\pi}" to define the return. Any particular reason why the more cumbersome "E_{s_0,a_0, ...}" was chosen here? Figure 1: Is there a reason you chose not to reference the ICLR 2018 paper? Also, why didn't you use the same illustration as they used, as the alternative one you provided does not seem more informative? It seems good to maintain consistency with the work you cite. L104 (eq (3)): I have hard time reading this expectation. What is the index i used for and what does it mean to condition on "m_i, s_i"? Please specify exactly the process for which we compute the return in Eq (3). Section 3, first paragraph. I really like the motivation provided here, especially the use of the advantage function. Consider adding a sentence on the fact that the discussion applies mostly in the case where we have good estimates for the Q-value and not in the early stages of learning. L148: "can not" -> "cannot" L154 (eq (5)): Just like Eq (3), this equation is overloaded and hard to interpret. First, how can an action be equal to a policy? I think you should replace "\pi_{*,t}" with "\pi_{*} (s_t)". Also, I am confused by the last part of the equation, when a_{master,t} is said to be part of the set "\{\pi_{nom}, \pi_{adv}}". What does that even mean? Please clarify. Figure (2): I believe it would be useful to explain what "m_i"'s correspond to in the proposed architecture. L164: "reported [13]" -> "reported in [13]". L215: Isn't the "actual advantage" the one defined under the true advantage? Consider using "observed advantage" instead. L223: I find it odd that you refer to [11] here instead of the paper that derived those bounds: Kakade & Langford (2002). Moreover, I believe you should add a more extensive discussion here on how one should interpret Proposition 1 and 2 + comparison to Kakade & Langford's bounds and a proper comparison is granted here. L267: I think it would be enough to say here "s_i + U(-a,a) for a>0" instead of the convoluted presentation in the submitted paper. L275: "Openai" -> OpenAI ----After author feedback---- I would like to thank the authors for the additional feedback, which provides clarifications for the major conceptual concerns I have raised. Moving some important clarifying remarks from the Supplemental material to the main paper is a great suggestion. Also, I agree that improving the accessibility of the paper to a wider audience is necessary and will increase the value of this work.